

# Rapid loss of phosphorus during early pedogenesis along a glacier retreat choronosequence, Gongga Mountain (SW China)

Yanhong Wu[1], Jun Zhou[1], Haijian Bing[1], Hongyang Sun[1] and Jipeng Wang[1,2]

[1] Key Laboratory of Mountain Surface Process and Ecological Regulation, Institute of Mountain Hazards and Environment, Chinese Academy of Sciences (CAS), Chengdu, China
[2] University of Chinese Academy of Sciences, CAS, Beijing, China

## ABSTRACT

The loss of phosphorus (P) during the early pedogenesis stage is important at the ecosystem level, and it also plays an important role in the global P cycle. The seasonal variation of total P (Pt) and its fractions along a young soil chronosequence (Hailuogou chronosequence) on the eastern slope of Gongga Mountain, SW China, was investigated based on the modified Hedley fractionation technique to understand P loss during the early pedogenesis stage. The results showed that the mineral P (mainly apatite) was the dominant fraction of Pt in the C horizon of the soil, and the seasonal difference in Pt and its fractions was insignificant. In the A horizon, Pt concentrations decreased markedly compared with those in the C horizon, and as the age of the soil increased, the inorganic P (Pi) significantly decreased and the organic P (Po) prominently increased. Seasonally, the P fractions exhibited various distributions in the A horizon. The variation of Pt and its fractions revealed that the P loss was rapid along the 120-year soil chronosequence. The P stocks in soils (0–30 cm) started to decrease at the 52 year site. And the P stock depletion reached almost 17.6% at the 120-year site. The loss of P from the soil of the Hailuogou chronosequence was mainly attributed to weathering, plant uptake, and transport by runoff. About 36% P loss was transported into plant biomass P at the 120 year site. The data obtained indicated that the glacier retreat chronosequence could be used to elucidate the fast rate of P loss during the early pedogenic stage.

## INTRODUCTION

Phosphorus (P) is one of the limiting nutrients in diverse natural habitats, including freshwater, marine, and terrestrial biomes (*Huang et al., 2013*; *Elser, 2012*), especially as nitrogen (N) deposition increases (*Elser et al., 2007*; *Craine & Jackson, 2010*; *Cramer, 2010*). Mountain regions, especially in some alpine and high-latitude ecosystems where the climate is cold and humid, are unique terrestrial ecosystems where P rather than N becomes

Corresponding author
Yanhong Wu, yhwu@imde.ac.cn

the main limiting nutrient (*Pérez et al., 2014*; *Seastedt & Vaccaro, 2001*; *Wassen et al., 2005*). The P limitation in terrestrial ecosystems results from the shortage of bio-available P in the soil (*Vitousek et al., 2010*). Due to the unique global P cycle (*Newman, 1995*; *Fillippelli, 2008*), the content of bio-available P in the soil decreases continuously during pedogenesis (*Walker & Syers, 1976*), resulting in P becoming the limiting nutrient and ecosystem regression in extreme situations (*Walder, Walker & Bardgett, 2004*; *Vitousek et al., 2010*).

In terrestrial ecosystems, the soil bio-available P is generally depleted in two ways. One is by P occlusion over time by biological and geochemical processes in which P is transformed into stable organic forms (*Turner et al., 2007*), which are difficult to mineralize (*Walker & Syers, 1976*). Another is the direct loss of bio-available P as well as other P fractions by plant uptake, soil erosion, runoff transport, etc. Compared with other ecosystems, the latter cause of P loss should be more significant in mountain ecosystems due to steep slopes, stronger runoff, and well-developed forests that take up more P. Furthermore, compared with occluding immobilization, the direct loss of P in mountain regions was almost completely neglected in former studies, and the loss rate and pathway were also far from clear.

Many previous studies found that the total P stock in soils would significantly decrease only after soils had been developed for thousands of years (e.g., *Galván-Tejada et al., 2014*; *Turner et al., 2013*; *Walker & Syers, 1976*). In recently developed soils (younger than one hundred years) where the occluding effect of P is not as remarkable as in well-developed soil (*Zhou et al., 2013*; *Prietzel et al., 2013*), the loss of P has not received enough attention so far. In the Hailuogou chronosequence, soil pH decreased sharply (*Zhou et al., 2013*) and vegetation succession developed fast (*Li & Xiong, 1995*) during the initial 120-year pedogenesis. Moreover, the precipitation in this area is rather abundant (mean annual precipitation: 1,947 mm) (*Wu et al., 2013*). These factors might lead to a rapid loss of P in the Hailuogou chronosequence. Therefore, the objective of this study is to know whether and how soil P was lost during the early pedogenic stage according to the seasonal variation of the concentrations, forms and stocks of P along this 120-year soil chronosequence. To achieve this objective, soil samples with 6 different ages were collected in the growing and non-growing seasons on the Hailuogou chronosequence. Soil P forms and relevant properties were measured and soil P loss was calculated based on these samples.

## MATERIALS AND METHODS

### Study area and samples collection

Hailuogou Glacier, located on the eastern slope of Gongga Mt., southwestern China (Fig. 1), has been retreating since 1890 (*Li et al., 2010*). A soil chronosequence has developed in the area exposed by the retreat that is approximately 2 km long, 50–200 m wide and has a 150 m altitude difference. The soil development on the chronosquence was described by *He & Tang (2008)*. According to the World Reference Base for Soil Classification (2006), the soils are grouped as Regosols. The mineral composition of the parent materials is similar in the retreat area, including plagioclase (28.5%), quartz (24.5%), biotite (13.7%), hornblende (13.0%), K-feldspar (8.6%) and apatite (1.6%) (*Yang et al., 2015*).

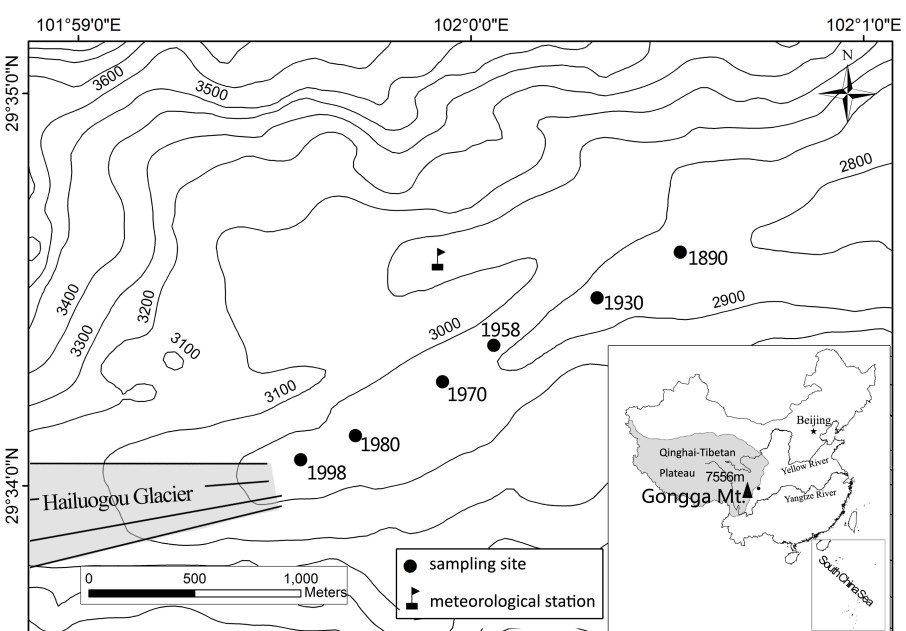

**Figure 1 Sketched map of Hailuogou Glacier retreat area and the sampling sites.**

A complete primary vegetation successional sequence has formed on the chronosequence, including (1) bare land, (2) *Salix rehderiana* C.K. Schneid.—*Hippophae rhamnoides* L.—*Populus purdomii* Rehder, (3) *Populus purdomii* Rehder, (4) *Abies fabri* (Mast.) *Craib*–*Picea brachytyla* (Franch.) E. Pritz., and (5) *Picea brachytyla* (Franch.) E. Pritz.—*Abies fabri* (Mast.) (*Li & Xiong, 1995*).

The climate on the Hailuogou chronosequence is controlled by the southeast monsoon with a mean annual temperature of 4.2 °C and a mean annual rainfall of 1,947 mm (*Wu et al., 2013*).

The soil samples were collected in December 2010 and July 2011 at the same plots in each site with different ages after the retreat of the glacier (Fig. 1). Three 2 × 2 m plots were created for each site, and the soil profiles were hand-dug at each plot. The soil profiles were divided into three horizons: the O, A and C horizons, except for the 12-year-old site, where the O and A horizon were absent.

At each site, the trunk, bark, latest-growth-year leaves and twig were collected and mixed to yield a composite sample from the dominant plants above the soil sampling sites. For each tree, a composite sample was collected from the upper and lower locations on the eastern, southern, western, and northern sides.

## Chemical analysis

A modified Hedley fraction extraction technique (*Tiessen & Moir, 1993*) was used to separate the P into eight fractions (Fig. 2). Two strips of an anion exchange membrane (BDH 551642S, 9 × 62 mm) were put into the tube. The anion exchange membrane strips were converted to bicarbonate before they were used in the first step. The supernatant of Step 1–4 was shaken for 16 h at 25 °C. Thereafter, the supernatant was centrifuged

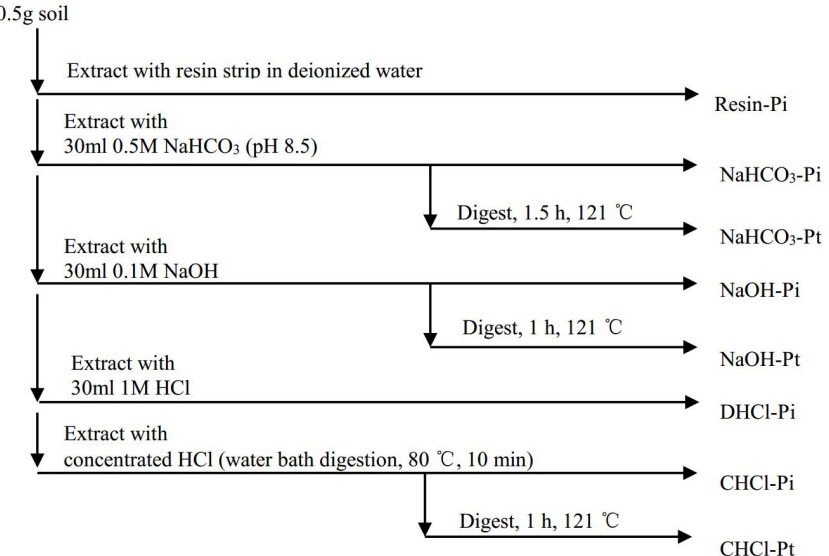

**Figure 2 The extraction procedure of P in soil (modified from *Tiessen & Moir, 1993*).**

at 25,000 × g for 10 min at 0 °C, and then passed through Millipore filters (pore size 0.45 μm). The concentrations of the organic P (Po) were calculated by the differences in the P concentrations between the undigested and the digested samples. The P concentrations were measured with a UV-V spectrophotometer (SHIMAZU UV 2450) using the phosphomolybdate blue method (*Murphy & Riley, 1962*). Blanks were mixed with the solvents to account for matrix interference during the extraction processes. The total P (Pt) concentration was measured using ICP-AES after subsamples were ground and digested with a microwave ($HNO_3$-$HClO_4$-$HF$).

The P fractions were defined as follows: (1) Resin-Pi (RPi, Resin-inorganic P) = exchangeable P, (2) $NaHCO_3$-P = labile Pi and Po, (3) NaOH-P = moderately bioavailable P, mainly Pi and Po tightly adsorbed and/or fixed by Al and Fe hydroxides, and P in humic and fulvic acids; (4) DHCl-Pi = Pi in primary minerals (mainly apatite); (5) CHCl-P = extractable P by concentrated HCl, refractory P (*Tiessen & Moir, 1993*).

Leaf, trunk, bark and twig were brushed clean and rinsed with deionized water before drying and grinding to a fine powder, respectively. The Pt concentrations of plant samples were measured by ICP-AES after microwave digestion ($HNO_3$-$H_2O_2$-$HF$). Values are expressed on a 60 °C dry-mass basis. The P concentration of plant samples was reported as the average concentration of leaf, trunk, bark and twig.

## Estimation of P loss

Because the largest thickness of A horizons was <12 cm in the Hailuogou chronosequence, P loss in the soils with a depth of 30 cm was evaluated. The P loss ($P_{LOSS}$) was calculated as following equation:

$$P_{LOSS} = \text{Stock-P}_{C30} - \text{Stock-P} \tag{1}$$

where Stock-$P_{C30}$ (kg/ha) is total P stock in the C horizon with a thickness of 30 cm and Stock-P is the sum of P stock in the O horizon (Stock-$P_O$) (kg/ha)and the surface mineral (Stock-$P_{AC30}$, 0–30 cm) (kg/ha) soils.

$$\text{Stock-}P_{C30} = C_{Pc} \times BD_c \times 30 \div 10 \qquad (2)$$

$$\text{Stock-}P = \text{Stock-}P_O + \text{Stock-}P_{AC30} \qquad (3)$$

$$\text{Stock-}P_O = C_{Po} \times BD_o \times D_o \div 10 \qquad (4)$$

where $C_{Po}$ (mg/kg), $BD_o$ (g/cm$^3$) and $D_o$ (cm) is the Pt concentration, bulk density and thickness in the O horizon, respectively.

$$\text{Stock-}P_{AC30} = (C_{PA} \times BD_A \times D_A + C_{PC} \times BD_C \times (30 - D_A)) \div 10 \qquad (5)$$

where $C_{PA}$ (mg/kg), $BD_A$ (g/cm$^3$) and $D_A$ (cm) is the Pt concentration, bulk density and thickness in the A horizon, respectively; $C_{PC}$ (mg/kg) and $BD_C$ (g/cm$^3$) is the Pt concentration and bulk density in the C horizon, respectively.

## RESULTS

The Pt concentrations in the C horizons across the six sites did not change significantly (Table 1) and were dominated by the DHCl-Pi, which accounted for 81–97% of the total P (Fig. 3). The seasonal variation in the concentrations of Pi, DHCl-Pi and Pt and its fractions in the C horizon was not significant (Fig. 3). The Po always only contributed a small part of Pt in the C horizon (Fig. 3).

Both of the Pt concentrations in the O and A horizons decreased with soil age (Table 1). At the 30–52 year sites, the Pt concentrations in the A horizons were slightly higher than that in the C horizons; while at the 80 and 120 year sites, the Pt concentrations in the O and A horizons were significantly lower than those in the C horizons (Table 1). The average of Pt concentration difference between the A and C horizons at the 80 and 120 year sites reached 381 mg/kg (Table 1).

For the P in the A horizons, the concentrations of Po increased continuously with the age of the soil, while the opposite trend was found for Pi (mainly DHCl-Pi) (Fig. 4). The concentrations of bio-available P (RPi + NaHCO$_3$-P) (*Tiessen & Moir, 1993*; *Wu et al., 2014*), as well as NaOH-P, slightly increased with the age of the soil. The significant seasonal difference in the P fractions was observed for NaOH-P and CHCl-P. The concentrations of NaOH-Po in the summer were approximately 100 mg/kg higher than those in the winter, which was mainly attributed to the increase in the concentration of NaOH-Po. On the contrary, the concentrations of CHCl-P in the winter were slightly lower than those in the summer, which was mainly related to the low concentration of CHCl-Pi.

## DISCUSSION

### Was P lost during the 120 years of pedogenesis?

The ratios of the Pt concentrations in the A horizon to that in the C horizon partly revealed the continuous loss of P with soil age. The ratio of $Pt_A/Pt_C$ was 0.62 and 0.74 at the

**Table 1** **Total P concentrations and soil properties along the Hailuogou chronosequence.**

| Age yrs | Dominant plants[*] | O thickness cm | A thickness cm | $pH_A$[*] | $SOM_A$[*] % | $Fe_{ox}$[*] mg/g | $Al_{ox}$[*] mg/g | $BD_O$[*] g/cm$^3$ | $BD_A$[*] g/cm$^3$ | $BD_C$[*] g/cm$^3$ | $Pt_O$ g/kg | $Pt_A$ g/kg | $Pt_C$ g/kg |
|---|---|---|---|---|---|---|---|---|---|---|---|---|---|
| 12 | *Astragalus adsurgens* Pall., *Hippophae rhamnoides* L., | – | – | 6.71 | 1.44 | 3.48 | 0.41 | – | – | 1.836 | – | – | 1,087 ± 94a |
| 30 | *Hippophae rhamnoides* L., *Populus purdomii* Rehder | 3 | 1 | 5.80 | 2.17 | 4.66 | 0.69 | 0.336 | 0.751 | 1.501 | 1,489 ± 236a | 1,193 ± 181a,b | 1,148 ± 92a |
| 40 | *Populus purdomii* Rehder (half-mature) | 5 | 1 | 5.44 | 4.40 | 4.21 | 1.22 | 0.308 | 0.657 | 1.543 | 1,331 ± 144a,b | 1,357 ± 147a | 1,304 ± 100a |
| 52 | *Abies fabri* (Mast.) Craib (half-mature) | 7 | 3 | 5.56 | 5.63 | 4.29 | 1.45 | 0.131 | 0.489 | 1.432 | 1,217 ± 95b | 1,147 ± 88b | 1,133 ± 54a |
| 80 | *Abies fabri* (Mast.), Craib *Picea brachytyla* (Franch.) E. Pritz. | 9 | 7 | 4.53 | 8.83 | 2.38 | 1.06 | 0.206 | 0.588 | 1.286 | 953 ± 26 c | 745 ± 101 c | 1,196 ± 178 a |
| 120 | *Picea brachytyla* (Franch.) E. Pritz., *Abies fabri* (Mast.) Craib | 11 | 9 | 4.79 | 8.69 | 4.74 | 1.95 | 0.118 | 0.598 | 1.334 | 900 ± 81c | 888 ± 221c | 1,199 ± 77a |

**Notes.**

O thickness and A thickness: the thickness of the O and A horizon. The thickness was reported using an average value of six profiles at each site (three profiles in 2010 and 2011, respectively).

[*] The data were from *Zhou et al. (2013)*. $BD_O$, $BD_A$ and $BD_C$: soil bulk density in the O, A and C horizon. The $pH_A$, $SOM_A$, $Fe_{ox}$ and $Al_{ox}$ represents pH, soil organic matter, amorphous Fe and Al in the A horizons, respectively.

$Pt_O$, $Pt_A$ and $Pt_C$: concentrations of total P in the O, A and C horizon. The P concentrations was reported using an average concentration in December 2010 and July 2011 (mean ± SD, $n = 6$).

Different letters indicate significantly different variables between different stages at the $p < 0.05$ level.

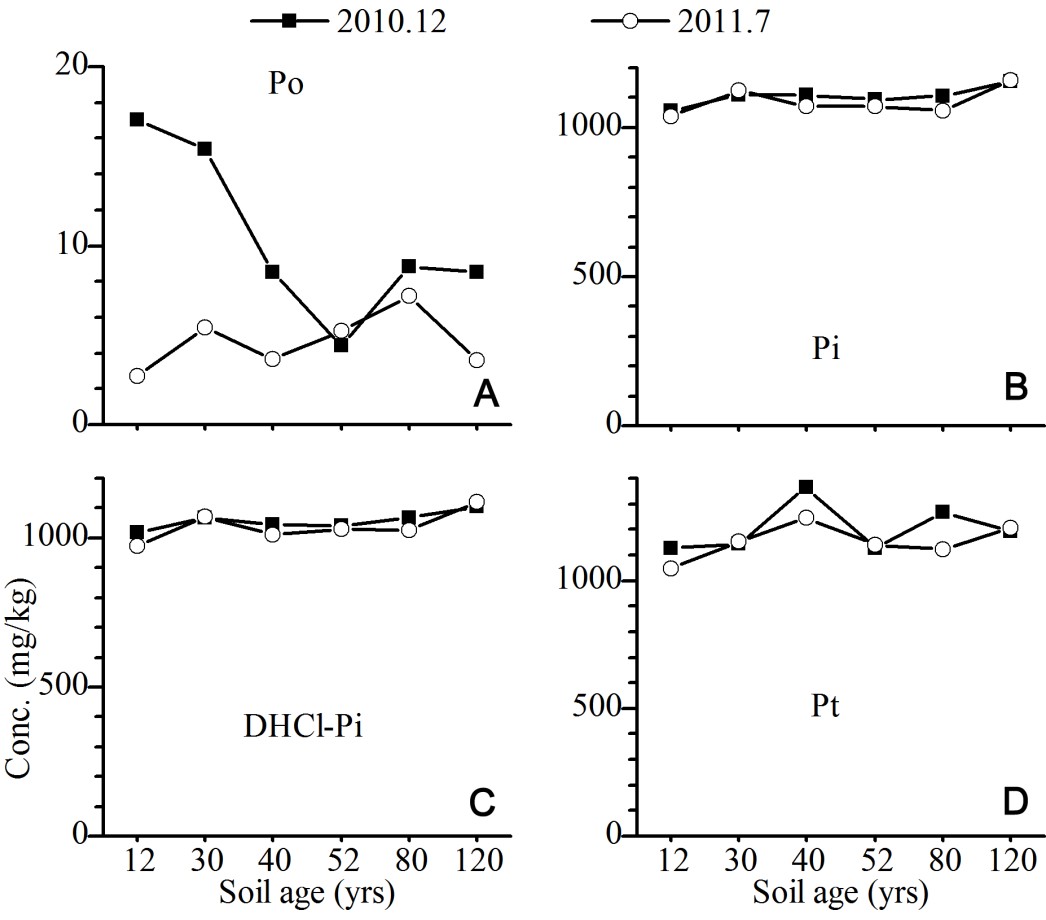

**Figure 3  Variations of the concentrations of total P and its fractions in C horizon of the Hailuogou chronosequence.**

80- and 120-year-old site, respectively. The ratios were much lower than those of the chronosequences of similar soil ages. There were no significant decreases of total P at the Morteratsch chronosequence (150 years) and the Damma chronosequence (120 years) in the Swiss Alps (*Egli et al., 2012*; *Prietzel et al., 2013*). Meanwhile, these ratios were also lower than or similar with those of several older chronosequences. The $Pt_A/Pt_C$ ratio at the 5,000-year-old site of the Franz Josef Chronosequence was approximately 72.7% (*Walker & Syers, 1976*). In a dune chronosequence in New Zealand, the ratio at the 370-year-old site was 59.5% (*Eger, Almond & Condron, 2011*). There was no obvious change in the total P at a 7,800-year-old chronosequence in Northern Sweden (*Vincent et al., 2013*). Lower $Pt_A/Pt_C$ ratios implied the severe loss of Pt in the soil under the climax community, and the loss rate was faster than other soil chronosequences.

The P stock in the O horizons and surface mineral soils (0–30 cm) directly showed the rapid loss of P on the Hailuogou chronosequence (Table 2). At the 30 and 40 year sites, P loss was not observed; while from the 52 to the 120 year site, the P loss increased with soil age. At the 120 year site, ∼17.6% Pt in the soils (thickness: 30 cm) was lost (Table 2). This is a relative high loss rate compared with other chronosequences with similar or greater

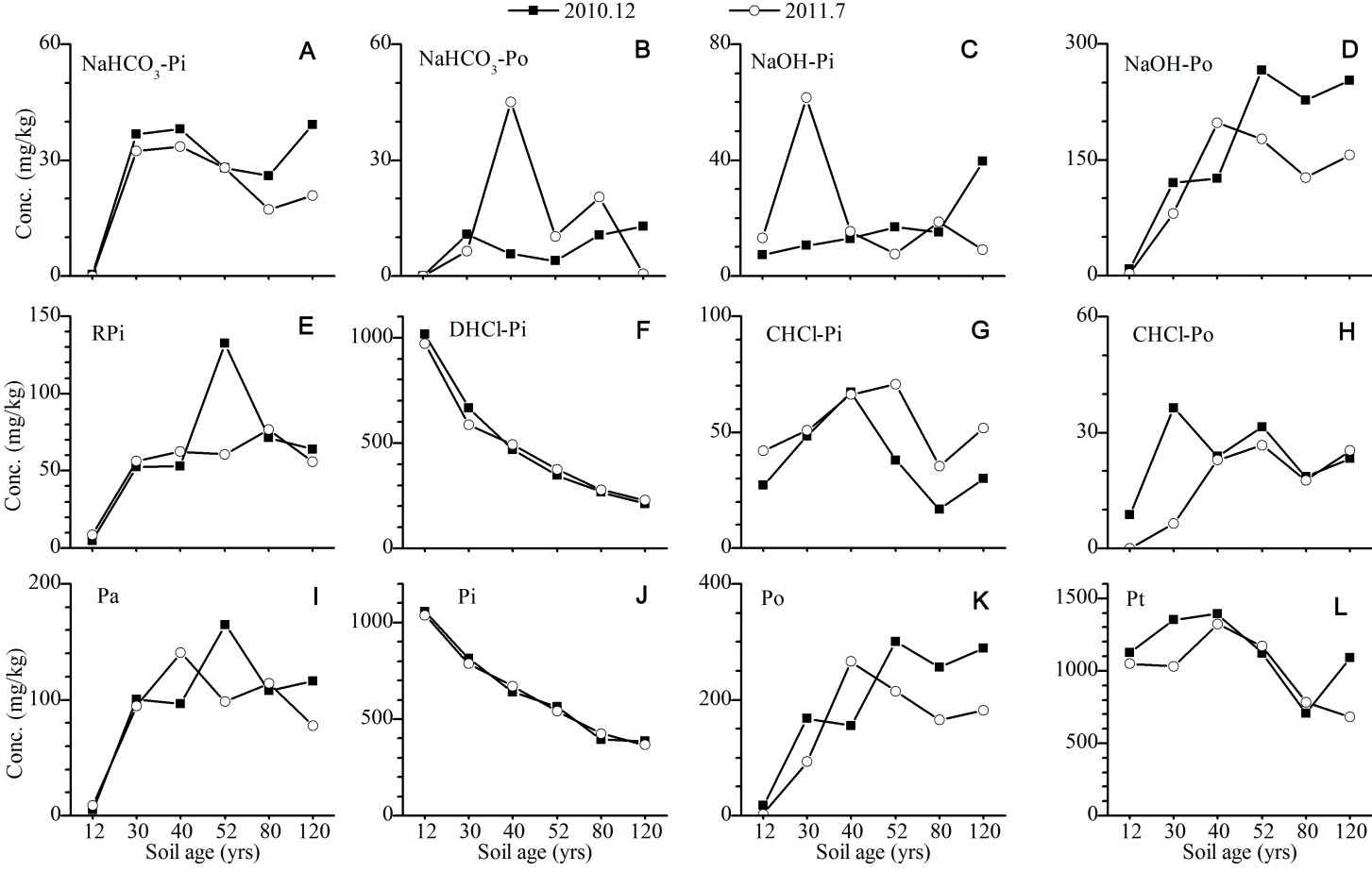

**Figure 4** **Variation of the concentrations of P fractions in the A horizons (The concentration of P in the 12 yr site was P in the surface sands (0–10 cm).** NaHCO$_3$-Pi and NaHCO$_3$-Po, inorganic and organic P extracted by NaHCO$_3$ solutions; NaOH-Pi and NaOH-Po, inorganic and organic P extracted by NaOH solutions; RPi, exchangeable P; DHCl-Pi, apatite P; CHCl-Pi and CHCl-Po, inorganic and organic P extracted by concentrated HCl; Pa, bioavailable P; Pi, inorganic P; Po, organic P; Pt, total P).

ages (Table 2). The P loss rate at the 120 year site on the Hailuogou chronosequence was ~7 times of that on the Rakata chronosequence, which was ~110 years old (Table 2). Although the P loss rate on the Hailuogou chronosequence was lower than those on some 'old' chronosequences (Table 2), the P loss could not be observed until these 'old' chronosequences had been developed for thousands of years. For example, *Galván-Tejada et al. (2014)* found that P stocks started to decline after 2,185 years of soil development on the Transmexican Volcanic Belt chronosequence. Moreover, *Turner et al. (2013)* found that the P stock (TP in soils < 2 mm) was 115.0, 194.2, 169.6 and 58.0 g P/m$^2$ at the 5, 1,000, 12,000 and 120,000 year site on the Franz Josef chronosequence, respectively, suggesting P stock did not decrease sharply at the initial stage of pedogenesis.

## The pathway of P loss along the soil chronosequence

During the early stage of pedogenesis, P could be released from the parent rocks into the soil by weathering. On the Hailuogou chronosequence, the degree and rate of weathering

Wu et al. (2015), *PeerJ*, DOI 10.7717/peerj.1377

**Table 2  Comparison of biomass, biomass P pool and P stock between the Hailuogou and other chronosequences.**

| No. | Age yrs | Site | Biomass[a] t/ha | $P_{plant}$[b] g/kg | Biomass P kg/ha | Stock-$P_O$ kg/ha | Stock-$P_{AC30}$ kg/ha | Stock-P kg/ha | Stock-$P_{C30}$ kg/ha | $P_{LOSS}$ Kg/ha | $P_{LOSS}$/ Stock-$P_{C30}$ % | Source |
|---|---|---|---|---|---|---|---|---|---|---|---|---|
|  | 12 | Hailuogou | 3.1 | 1.6 | 5.0 | N.D. | 5,988 | 5,988 | 5,988 | 0 | 0.0 | This study |
|  | 30 | Hailuogou | 48.9 | 1.2 | 57.5 | 150 | 5,088 | 5,238 | 5,170 | −67 | −1.3 | This study |
| 1 | 40 | Hailuogou | 110.8 | 0.9 | 99.7 | 205 | 5,924 | 6,129 | 6,036 | −93 | −1.5 | This study |
|  | 52 | Hailuogou | 184.7 | 0.9 | 164.9 | 112 | 4,548 | 4,659 | 4,866 | 207 | 4.3 | This study |
|  | 80 | Hailuogou | 308.0 | 0.7 | 225.2 | 177 | 3,845 | 4,021 | 4,615 | 593 | 12.9 | This study |
|  | 120 | Hailuogou | 382.3 | 0.8 | 303.9 | 117 | 3,836 | 3,953 | 4,798 | 845 | 17.6 | This study |
| 2 | 40 | Maluxa |  |  |  | 23 | 910 | 933 | 816 | −117 | −14.3 | *Celi et al., 2013* |
| 3 | 98 | Morteratsch |  |  |  | N.D. | 1,122 | 1,122 | 1,046 | −75 | −7.2 | *Egli et al., 2012* |
| 4 | 110 | Rakata |  |  |  | N.D. | 1,129 | 1,129 | 1,157 | 28 | 2.4 | *Schlesinger et al., 1998* |
| 5 | 6,500 | S. Westland |  |  |  | N.D. | 487 | 487 | 1,822 | 1,335 | 73.3 | *Eger, Almond & Condron, 2011* |
| 6 | ∼3,200 | Cooloola | $P_{LOSS}$ is the difference of P stocks between 400 and ∼3,200 yrs site. 400 yrs: 260 kg P/ha, ∼3,200 yrs: 119 kg P/ha. |  |  |  |  |  |  | 141 | 54.2 | *Chen et al., 2015* |
| 7 | ∼6,500 | Jurien Bay | $P_{LOSS}$ is the difference of P stocks between <100 and ∼6,500 yrs site. <100 yrs: 3,843 kg P/ha, ∼6,500 yrs: 1,948 kg P/ha. |  |  |  |  |  |  | 1,895 | 49.3 | *Turner & Laliberté, 2015* |
| 8 | 120,000 | Franz Josef | $P_{LOSS}$ is the difference of P stocks between 5 and $1.2 \times 10^5$ yrs site. 5 yrs: 1,150 kg P/ha, $1.2 \times 10^5$ yrs: 580 kg P/ha. |  |  |  |  |  |  | 570 | 49.6 | *Turner et al., 2013* |
| 9 | >100,000 | Transmexican Volcanic Belt | P stocks declined after 2,185 yrs of soil development. |  |  |  |  |  |  |  |  | *Galván-Tejada et al., 2014* |

**Notes.**

[a] Biomass was the sum of above and below ground calculated according to *Luo et al. (2004)*.

[b] The P concentration was the average concentration of leaf, trunk, bark and twig.

Stock-$P_O$: P stock in the O layer.

Stock-$P_{AC30}$: total P stock in the surface mineral soils (depth: 0–30 cm).

Stock-P: Stock-$P_O$ + Stock-$P_{AC30}$

Stock-$P_{C30}$: total P stock in the C horizon with a thickness of 30 cm.

$P_{LOSS}$: Stock-$P_{C30}$−Stock-P.

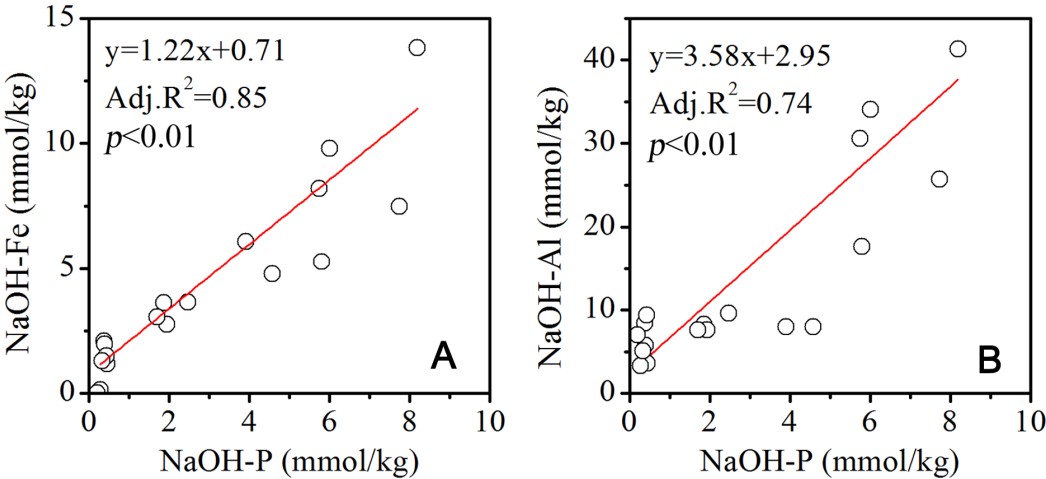

**Figure 5 Correlation between P and Fe, Al in the NaOH extracted solution.**

changed with soil age (*Zhou, 2014*). The weathering rate at the 120-year-old site was 111 $cmol_c/m^2$ yr, which was higher than those in Alps and some tropical zones (*Egli, Fitze & Mirabella, 2001*; *Taylor & Blum, 1995*). The rapid weathering process could lead to the dissolution of apatite bound to carbonates, biotite, hornblende, plagioclase and other silicate minerals. The depletion factor of soil P on the Hailuogou chronosequence demonstrated that only approximately 22% of the original apatite was remained after 120 years of weathering and pedogenesis (*Zhou, 2014*). The rapid dissolution of apatite during the weathering process should be ascribed to the fine materials that resulted from strong glaciation and freeze-thaw processes, adequate moisture and sharp decrease of pH (*Zhou et al., 2013*) due to rapid vegetation succession.

P released from minerals as phosphate was available for plant assimilation. Along the Hailuogou chronosequence, the primary vegetation succession was quickly established with the weathering and pedogenesis. The increase of biomass, as well as P in biomass, was also prominent along the chronosequence (Table 2). The pool of P in biomass in the forest at the 120 year site was 303.9 kg/ha, which accounted for ∼36% of P loss (Table 2). Therefore, plant uptake was a significant pathway of P loss from the soil. Moreover, the lower concentrations of bio-available P in July (Fig. 4), the growing season, confirmed that plant uptake accelerated the loss of P from the soil.

Besides of the P assimilated by plant, about 540 kg P per ha was lost on the chronosequence. This suggested that in addition to plant uptake, there should be other paths for P loss from the soil on the Hailuogou chronosequence.

As P was released from parent rocks, Fe, Al, Ca and other metal ions were released as well during weathering and pedogenesis. Fe and Al ions were likely to form hydroxides. In addition to being assimilated by plants, the released phosphate tended to be adsorbed onto the surface of Fe and Al hydroxides. In the NaOH extracted solutions, the P concentrations were significantly positively related with Fe and Al concentrations (Fig. 5). This result implied that the P extracted by NaOH was from Fe and Al hydroxides that bound P, which accounts for more than 30% of the Pt at the 52, 80- and 120-year-old site in December,

while it was a little lower in July. A previous study in dark coniferous forests showed that the P bound by Al and Fe was the major fraction of soil P (*Wood, Bormann & Voigt, 1984*). *Kaňa & Kopáček (2005)* confirmed that the adsorption capacity of Fe and Al hydroxides was the dominant factor controlling the transport of soil P to the water body. *Šantrůčková et al. (2004)* demonstrated that the lower acidity of the forest soil drove the transformation of P and changed the P loss rate. As the vegetation established and evolved on the Hailuogou chronosequence, the soil pH decreased with the soil age (*Zhou et al., 2013*). In the acidified soils, especially at the 80- and 120-year-old sites, the P bound by Fe and Al was potentially transported by runoff. The annual precipitation was concentrated in the summer (*Wu et al., 2013*). The P bound by Fe and Al could be more significantly discharged by runoff in the summer than in the winter, which could be interpreted as the reason for the lower contribution of NaOH-P to the Pt in July than in December. Meanwhile, the elevation difference of 150 m along less than 2 km of the Hailuogou chronosequence, led to the strong erosion and great loss of soil P. The mean concentration of Pt in the Hailuogou glacier waters was $0.049 \pm 0.014$ mg/l, according to the Alpine Ecosystem Observation and Experiment Station of Mt. Gongga, Chinese Academy of Sciences (2002–2005). The annual runoff was 307.1 $m^3$/yr ha in the Hailuogou glacier watershed (*Li et al., 2004*). The P discharge rate was calculated as 1.5 kg P/yr ha.

## CONCLUSION

During the early stage of pedogenesis at the Hailuogou chronosequence, P was rapidly lost from the soil. The P loss could be observed after 52 years of deglaciation, and the loss reached 12.9% and 17.6% at the 80- and 120-year-old site, respectively. The fast loss of P from the soil could be attributed to the higher weathering rate, the large amount of plant uptake and transport by runoff.

### Funding
This work was supported by the National Natural Science Foundation of China (Grant No.: 41272200). The funder had no role in study design, data collection and analysis, decision to publish, or preparation of the manuscript.

### Grant Disclosures
The following grant information was disclosed by the authors:
National Natural Science Foundation of China: 41272200.

### Competing Interests
The authors declare there are no competing interests.

### Author Contributions
- Yanhong Wu conceived and designed the experiments, performed the experiments, analyzed the data, contributed reagents/materials/analysis tools, wrote the paper, prepared figures and/or tables, reviewed drafts of the paper.

- Jun Zhou performed the experiments, analyzed the data, prepared figures and/or tables, reviewed drafts of the paper.
- Haijian Bing and Hongyang Sun performed the experiments, contributed reagents/materials/analysis tools, reviewed drafts of the paper.
- Jipeng Wang performed the experiments, contributed reagents/materials/analysis tools, prepared figures and/or tables, reviewed drafts of the paper.

## Supplemental Information

Supplemental information for this article can be found online at http://dx.doi.org/10.7717/peerj.1377#supplemental-information.

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
