# Peer review of "Rapid loss of phosphorus during early pedogenesis along a glacier retreat choronosequence, Gongga Mountain (SW China)"

_PeerJ, doi:10.7717/peerj.1377_

## Round 0.1 · original submission · Major Revisions

The paper presents an interesting study on total P loss in a 120 years chronosequence. The authors need to clarify that part of the research which was published previously in Geoderma (2013) and stated the new contribution of this paper. Reviewer 1 was concerned that the authors did not take into account all the P fractions in the O horizons.

·

Basic reporting

The submission satisfies PeerJ's basic reporting guidelines. However, the authors did not include page numbers or line numbers on their manuscript. Line numbers in particular make it easier for both referees and authors to specifically point out sections of text during the review process.

Experimental design

The submission describes original research with a clearly defined research question. However, methods are insufficiently described.

It appears there are six research sites according to the map in Figure 1 and subsequent data figures, but the description of the vegetation sequence lists only five sites.

The authors do not provide any information on how extractants (supernatants) were digested, nor is it clear whether the soil was analyzed for total P either before or after fractionation. As written, it appear the authors are summing P extracted during the fractionation process without incorporating the final step (as described in Tiessen and Moir) of estimating total P in soil either by digestion or by fusion (see Smith and Bain 1982).

It does not appear the authors analyzed P in the O horizon, which would increase estimates of total P at the older five sites where an O horizon is present, and alter the disparity in total P between the youngest site (1), where no O horizon is present, and older sites.

Validity of the findings

Based on the methods as currently described, the authors are likely underestimating total soil P at their research sites for two reasons. First, they do not include P in the O horizon, which could be a substantial pool of organic P (Po). Second, the authors do not appear to estimate total P in the soil, either before or after sequential fractionation. The difference between total P in the soil and the sum of all previously extracted P represents the most occluded P pool. I consider it highly unlikely that all soil P was extracted during the sequential fractionation as currently described.

If the authors are indeed underestimating the total soil P pool, this would greatly alter their findings re: patterns in both total P and individual P fractions over time.

If samples were archived, I recommend the authors analyze soil samples for total P using the fusion procedure outlined by Smith and Bain (1982) in the journal Communications in Soil Science and Plant Analysis.

Reviewer 2 ·

Basic reporting

Figures. Perhaps Fig. 1 could have also been plotted using a DEM of the area (if available). Figures were included in the .doc file, thus it was not possible to evaluate them as unbound files. The authors should make sure that the resolution is high enough for the journal standards.

Literature. Especially in the introduction, I would suggest the authors to consider the following literature as well:

N and P fixation and deposition:

Knelman, J., et al. 2012. Bacterial community structure and function change in association with colonizer plants during early primary succession in a glacier forefield.

Perez, C.A., et al. 2014. Ecosystem development in short‐term postglacial chronosequences: N and P limitation in glacier forelands from Santa Inés Island, Magellan Strait

P transformations:

Chapin, F.S., et al. 1994. Mechanisms of Primary Succession Following Deglaciation at Glacier Bay, Alaska

Turner, B.L., et al. 2007. Soil Organic Phosphorus Transformations During Pedogenesis

Evidences of apatite and P-bearing minerals as sources for inorganic P:

Mavris, C., et al. 2012. Weathering and mineralogical evolution in a high Alpine soil chronosequence: A combined approach using SEM–EDX, cathodoluminescence and Nomarski DIC microscopy.

Wongfun, N., et al. 2013. Effect of water regime and vegetation on initial granite weathering in a glacier forefield: Evidences from CL, SEM, and Nomarski DIC microscopy

Raw data. On the raw data Excel file there is a field named 冰退时间. Most certainly this is related to the age of the sampling site, however the entire content of the manuscript must be written in English. Please translate it to English, in order to clarify your message to the reader.

Experimental design

Overall, the design is sound and properly carried out. However the manuscript lacks two basic tables to better address the authors’ research questions, to be preferentially included in the Materials and Methods section:
- The first one should report the geographic sampling context (i.e. mean annual temperature, slope, etc.). The work from Zhou et al. (2013) would be a good example to present and cite, and it would add a better vision of the area to the reader.
- The second table should report the soil physical properties and a detailed soil profile description for the sampled sites. If no original samplings were performed, data from He and Tang (2008) and Zhou et al. (2013) would be worth displaying in a table form, describing in detail soil depth, description of the soil horizons, grain size, content of SOM, and so forth.
The above tables become crucial into the contextualisation of the findings.

Validity of the findings

The findings are sound and consistent with available literature, but the conclusions should be stated a bit more clearly and extensively.

Additional comments

It is with great pleasure that I have reviewed the manuscript PeerJ5636, entitled:
‘Rapid loss of phosphorus during early pedogenesis along a glacier retreat choronosequence, Gongga Mountain (SW China)’.
The manuscript is concise and to the point. The writing style of the authors is clear, in correct English, and typically with well-addressed sentences. The approach is scientifically sound, in line with existing literature, and the work displays very well the P depletion (and the characterisation of the different P types) in the selective proglacial field.
However, the previously (and following) amendments must be performed before the acceptance of the paper:

Title
- please amend ‘choronosequence’ into ‘chronosequence’

Abstract
- Line 6: it would be clearer by changing ‘mineral P (mainly apatite)’ into ‘inorganic, mineral-derived P (mainly apatite-derived)’. Also, when you specify 'mainly apatite', do you hypothesise another source for inorganic P? If yes, please specify.
- Line 17: instead of ‘glacier retreat chronosequence’ it addresses better ‘proglacial chronosequence’
- Line 19: ‘early pedogenic stages’

Introduction
- Lines 49-55: the transition from the gap of knowledge to the aim of the work is not totally clear, the sentences look rather disconnected. Please reformulate accordingly.

Materials and Methods
- Line 64-67: was apatite detected with means of XRD? If yes, please quantify it. If not, please report so.
- Lines 115-118: please provide a solid reference (i.e. paper, database, Ministry) for the Chinese geological reference materials.

Discussion
- Lines 180-182: why is there a space between the two paragraphs? If no need, please remove it.
- Line 186: please change ‘montane’ with ‘mountain’
- Line 204: did you mean ‘Nezat et al, 2004’? If yes, please amend.

---

## Round 0.2 · accepted · Accept

The paper has been revised and ready for publications, however there are a few typos and a mistake in the equation, which should be fixed before publication:

Eq. (2) Should be CPc x BDc x 30 /10
Should be no "O" subscript as it refers to the C horizon
Page 9, Line 2, Fe and Al ions were "apt"
Not sure what "apt" means